# Virulence and Infectivity of UC, MD, and L Strains of Infectious Hematopoietic Necrosis Virus (IHNV) in Four Populations of Columbia River Basin Chinook Salmon

**DOI:** 10.3390/v13040701

**Published:** 2021-04-18

**Authors:** Daniel G. Hernandez, William Brown, Kerry A. Naish, Gael Kurath

**Affiliations:** 1Western Fisheries Research Center, United States Geological Survey, Seattle, WA 98115, USA; Hernandez.Daniel.LLC@gmail.com; 2School of Aquatic and Fishery Sciences, University of Washington, Seattle, WA 98195, USA; knaish@uw.edu; 3Department of Statistics, University of Washington, Seattle, WA 98195, USA; brownw@uw.edu

**Keywords:** virulence, infectivity, infectious hematopoietic necrosis virus, IHNV, infectious dose, Columbia River Basin, Chinook salmon, intraspecies variation

## Abstract

Infectious Hematopoietic Necrosis Virus (IHNV) infects juvenile salmonid fish in conservation hatcheries and aquaculture facilities, and in some cases, causes lethal disease. This study assesses intra-specific variation in the IHNV susceptibility of Chinook salmon (*Oncorhynchus tshawytscha*) in the Columbia River Basin (CRB), in the northwestern United States. The virulence and infectivity of IHNV strains from three divergent virus genogroups are measured in four Chinook salmon populations, including spring-run and fall-run fish from the lower or upper regions of the CRB. Following controlled laboratory exposures, our results show that the positive control L strain had significantly higher virulence, and the UC and MD strains that predominate in the CRB had equivalently low virulence, consistent with field observations. By several experimental measures, there was little variation in host susceptibility to infection or disease. However, a small number of exceptions suggested that the lower CRB spring-run Chinook salmon population may be less susceptible than other populations tested. The UC and MD viruses did not differ in infectivity, indicating that the observed asymmetric field prevalence in which IHNV detected in CRB Chinook salmon is 83% UC and 17% MD is not due to the UC virus being more infectious. Overall, we report little intra-species variation in CRB Chinook salmon susceptibility to UC or MD IHNV infection or disease, and suggest that other factors may instead influence the ecology of IHNV in the CRB.

## 1. Introduction

The aquatic rhabdoviral pathogen infectious hematopoietic necrosis virus (IHNV) infects many species of Pacific salmonid fish. In the multi-host ecosystems of western North American rivers, IHNV has diverged into three major virus genogroups that vary in host-specific virulence for three major host species: Chinook salmon (*Oncorhynchus tshawytscha*), sockeye salmon (*O. nerka*), and steelhead/rainbow trout (both *O. mykiss*) [1,2]. In the large Columbia River Basin (CRB) of the US Pacific Northwest, the most abundant host is Chinook salmon, and recent studies suggest that subclinical infection of Chinook salmon with IHNV may be contributing to the transmission and maintenance of the virus across the watershed [3,4]. Chinook salmon occur in the CRB as numerous distinct sub-populations and exhibit dramatic variation in life-history phenotypes. The dominant Chinook salmon life-history phenotypes in the CRB are known as spring-run and fall-run Chinook salmon, which differ in many ecologically important traits, including spatial ranges (Figure 1) and seasonal timing of both juvenile seaward migration and adult upstream migration for spawning [5,6]. Phylogenetic analyses of CRB Chinook salmon by various methods have revealed three or four clearly distinct genetic lineages that vary in geographic distribution and life-history phenotypes [6,7,8,9]. To date, variation in host specificity of IHNV has been defined at the host species level, but potential differences in the interactions of IHNV with different life-history phenotypes or phylogenetic lineages of CRB Chinook salmon have not been explored. The goal of this investigation was to test for intraspecies variation in the susceptibility of CRB Chinook salmon populations to infection and disease following exposure to distinct strains of IHNV. Here, we measured the virulence and infectivity of three IHNV strains representing relevant virus genogroups, in four diverse populations of CRB Chinook salmon, including spring- and fall-run populations from the upper and lower regions of the watershed. By combining results of controlled laboratory exposures with field surveillance data, we assessed factors of both the host and virus, leading to observed patterns of IHNV infection and virulence in CRB Chinook salmon. Together, this investigation aimed to better understand the contributions of distinct Chinook salmon populations to the ecology and epidemiology of IHNV across the Columbia River Watershed.

IHNV is an enveloped, single-stranded, negative-sense RNA virus (species *Salmonid novirhabdovirus*) belonging to the taxonomic family *Rhabdoviridae*, in the genus *Novirhabdovirus* [11,12,13]. IHNV can cause high mortality in juvenile salmonids in conservation hatcheries and commercial aquaculture facilities, and disease impacts have been well documented in rainbow trout and steelhead trout (freshwater and anadromous forms *O. mykiss*), sockeye salmon, and Chinook salmon [11,12]. Genetic typing and phylogenetic analyses of virus field isolates have identified three major genogroups of IHNV in North America designated U (upper), M (middle), and L (lower) for their relative geographic distributions [1,14,15]. Distinct patterns of host-specific virulence have been observed for these genogroups, with infection frequency and disease outbreaks due mostly to L viruses in juvenile Chinook salmon, U viruses in juvenile sockeye salmon, and M viruses in juvenile rainbow and steelhead trout. Geographically, disease outbreaks in Chinook salmon occur mostly in California and southern Oregon within the L genogroup range [16], outbreaks in sockeye salmon occur mostly in the U genogroup range in northern coastal watersheds from Oregon to Alaska [1], and outbreaks in steelhead/rainbow trout are mostly within the CRB [15,17] where the U and M genogroup ranges overlap [14,15]. 

As the most productive Pacific salmon watershed in the contiguous United States [6], the CRB supports the three predominant host species for IHNV, with Chinook salmon by far most abundant, followed by steelhead trout, and smaller populations of sockeye salmon [18]. The prevalence of IHNV across the CRB and coastal watersheds of Washington and Oregon is reported as 29.1% in steelhead trout, 21.9% in sockeye salmon, and 20.1% in Chinook salmon [4]. In the CRB, M viruses occur as specialists in steelhead and rainbow trout, with a low frequency of detections in other host species, including Chinook salmon [15]. M viruses in the CRB are nearly all in the MD subgroup of the M genogroup, and they periodically cause epidemic disease in steelhead trout [15]. In contrast, U genogroup viruses in the CRB are detected frequently in both Chinook salmon and steelhead trout, as well as in small numbers of sockeye salmon [15]. This is a unique host association pattern for U genogroup viruses, which occur as sockeye salmon specialists outside the CRB. A detailed phylogenetic study of U genogroup viruses revealed that a novel U subgroup has evolved in the CRB, designated UC, which is associated with this more generalist host pattern infecting all three predominant IHNV host species [17]. Of relevance to the epidemiology and overall ecology of IHNV in the CRB is the high prevalence of the virus in the abundant CRB Chinook salmon, without the disease impacts observed in CRB steelhead trout, coastal sockeye salmon, and Chinook salmon in California.

The virulence of IHNV in Chinook salmon has been previously characterized in experimental challenges using L virus strains in fish from southern Oregon and northern California [19,20,21]. Other early experimental studies have compared the virulence of different IHNV isolates, now known to be L, U, or M strains, in Chinook salmon from Alaska or the Columbia River Watershed [22,23]. These comparative studies in Chinook salmon found that L strains of IHNV were more virulent than the U and M viruses. Moreover, one early study reported that Chinook salmon from a Columbia River population was significantly more susceptible to mortality after an L virus challenge than two Alaskan Chinook salmon populations, providing the first demonstration of intraspecific variation in IHNV susceptibility of Chinook salmon [22]. These early studies all measured virulence in terms of mortality after the immersion challenge, but did not assess infection.

In a recent study by Hernandez et al. [3], both infectivity and virulence of five IHNV isolates, including L, U, and M strains, were defined in two populations of Chinook salmon from the CRB or from the Puget Sound region (not within the CRB). After exposure to a high dosage (2 × 10^5^ pfu mL^−1^) of each IHNV strain by batch immersion, only the L strain caused significant mortality compared with mock-challenged groups. Infection frequency seven days after challenge was significantly higher for L than for some U and M strains, and there was a trend of higher infectivity for U strains relative to M strains. In addition, CRB Chinook salmon appeared to be generally more susceptible to infection than the Puget Sound population [3]. These observations suggested the possibility of intraspecific variation in Chinook salmon susceptibility to IHNV, prompting the current investigation.

Here we conducted controlled laboratory exposures using diverse populations of CRB Chinook salmon to elucidate possible intraspecific variation in host susceptibility to three strains of IHNV. In Waples et al. [7], four major phylogenetic lineages were described for Chinook salmon of the CRB. Therefore, we selected four experimental host populations of CRB Chinook salmon to represent each of these four phylogenetic lineages (Table 1a). Geographically, two of these Chinook salmon populations were obtained from the upper Columbia River (East of the Cascade Mountain Range) and two from the lower Columbia River (west of the Cascade Mountain Range) (Figure 1). In each geographic region, we included one spring-run and one fall-run population [7,9]. These distinct adult life-history phenotypes generally correspond to distinct juvenile life-history phenotypes observed in freshwater rearing [6], known as stream-type and ocean-type Chinook salmon, respectively [24,25]. We refer to these four populations hereafter as upper spring-run (Up-Spring), upper fall-run (Up-Fall), lower spring-run (Low-Spring), and lower fall-run (Low-Fall) Chinook salmon (Figure 1). The three virus strains used in the challenge experiments were chosen to represent the most dominant genetic types of the UC and MD subgroups that occur in the CRB, as well as an L genogroup strain from California as a positive control known to have high virulence in Chinook salmon.

In all experimental challenge studies, mortality and infection outcomes were assessed from both the virus and host perspectives. Variation in IHNV virulence and host susceptibility to disease was assessed by monitoring mortality over a 30-day time course following immersion exposure to two different doses of each virus strain. Variation in virus infectivity and host susceptibility to infection was assessed by quantifying infection frequency and magnitude of viral load three days after exposure to the virus. For infectivity assays, each experimental host population was exposed by immersion to four different concentrations of virus, and the dosages needed to infect 50 percent of each population (ID_50_) with each virus strain were determined as a metric of virus infectivity. The influence of the host population, viral strain, and exposure dose on the outcomes of infection (infection status and viral load) were also evaluated using fixed-effects logistic regression. Finally, we compared the results of our controlled laboratory studies with field surveillance data to more comprehensively understand the host-pathogen interactions of CRB Chinook salmon with UC and MD IHNV. To do this, we assessed field patterns of IHNV infection in different life-history phenotypes of CRB Chinook salmon from virus surveillance data and genetic typing information. This multifaceted investigation is the first to assess intraspecies variation in susceptibility of diverse populations of CRB Chinook salmon to infectivity and mortality caused by different strains (UC, MD, and L) of IHNV.

## 2. Materials and Methods

### 2.1. Experimental Challenges: Host Populations

Chinook salmon of the Methow River (from Winthrop National Fish Hatchery, United States Fish and Wildlife Service) and North Santiam River (from Marion Forks Hatchery, Oregon Department of Fish and Wildlife) were selected to represent upper and lower CRB spring-run populations (Table 1a), designated hereafter as Up-Spring and Low-Spring, respectively (populations 28 and 8 in Narum et al. [9]). Chinook salmon of the Hanford Reach (from Priest Rapids Hatchery, Washington Department of Fish and Wildlife) and Cowlitz River (from Cowlitz Salmon Hatchery, Washington Department of Fish and Wildlife) were selected as upper and lower CRB fall-run populations, designated hereafter as Up-Fall and Low-Fall, respectively (populations 14 and 1 in Narum et al. [9]). In each case, the identity of the eggs as representing the specific population reported in Narum et al. [9] was verified by the hatchery managers and/or agency fish geneticists. A total of 2000 eyed eggs (developing embryo), sourced from a minimum of 12 parental spawning pairs tested negative for IHNV, were obtained from each of the four populations. All Chinook salmon eggs were disinfected with ARGENTYNE^®^ buffered povidone iodine (Argent Aquaculture LLC, Redmond, WA, USA) using standard biosecurity procedures recommended for control of IHNV [26], incubated and hatched at the U.S. Geological Survey (USGS) Western Fisheries Research Center (WFRC) laboratory in Seattle, WA, where they were reared to approximately 1 g at a constant temperature of 10 °C. Juvenile Chinook salmon were fed a semi-moist pellet diet (Bio-Oregon) at a rate of 1.0–2.0% body weight per day. All fish rearing and experimental exposures were conducted at the USGS WFRC using single-pass, flow-through, sand-filtered and UV-treated freshwater from Lake Washington.

### 2.2. Experimental Challenges: Virus Exposures

Three viral strains were included in this investigation (Table 1b). Strain RB1, representing the UC subgroup, is virulent in sockeye salmon [27], and has the most common UC sequence type detected in the CRB (type mG001U) [14,15]. Strain QTS07, representing the MD subgroup, has high virulence in steelhead trout [28], and has the most common MD IHNV sequence type (type mG110M) detected throughout the CRB [15]. Although the L genogroup of IHNV does not occur in the CRB, the California L genogroup strain FR0031 was included as a positive control known to have high virulence and infectivity in juvenile Chinook salmon [3,21]. Controlled laboratory challenges were performed on juvenile Chinook salmon at an average weight of 1 g. Each host population was exposed to L, UC, and MD genogroups strains of IHNV (Table 1b), as well as to a virus-free media treatment (negative control). All viral challenges were conducted at a constant water temperature of 10 °C selected as an average value for general conditions observed throughout the CRB [6].

To assess IHNV virulence, triplicate groups of twenty fish were exposed to each viral strain by static immersion for 1 h in 1 L of water containing the virus at a concentration either of 2 × 10^3^ plaque-forming units (PFU) mL^−1^ (referred to as ‘moderate dose’) or 2 × 10^5^ PFU mL^−1^ (referred to as ‘high dose’). These doses allow us to assess response to a standard high dose used in previous work [27,28], and a more moderate dose to reveal differences that may be saturated at the high dose. After immersion challenge, water flow was resumed, and each tank filled to a final volume of 5 L. These triplicate groups were monitored daily over the course of 30 days for mortality. All deceased fish were removed daily. Cumulative percent survival (CPS) for each combination of the host population and viral strain was calculated as the average CPS across triplicate groups for each combination.

At the end of the 30-day observation period, all surviving fish were euthanized, and eight fish from each high dose treatment group were saved for a preliminary analysis of viral persistence. These fish were stored at −80 °C for later RNA extraction and quantification of viral RNA.

To characterize the infectivity of L, UC, and MD virus types in diverse populations of CRB Chinook salmon, twenty additional fish from each experimental population were exposed to each virus strain at four virus concentrations: 2 × 10^2^, 2 × 10^3^, 2 × 10^4^, and 2 × 10^5^ PFU mL^−1^. These doses span from low to high infection probabilities for IHNV, as observed previously [29]. Viral immersion challenges were conducted for 1 h as described above. Following the 1 h rinse, 8 of the 20 juvenile fish in each viral treatment were placed into individual 1 L beakers containing 400 mL of static water maintained at a constant temperature of 10 °C by circulating temperature-controlled water around the beakers. The remaining 12 fish from each virus treatment were euthanized using buffered Tricaine Methanesulfonate (Syndel, Ferndale, WA, USA) at a concentration of 240 mg L^−1^. Three days post exposure (dpe), each set of eight fish per treatment group were euthanized and individually stored at −80 °C for RNA extraction.

### 2.3. Measures of Viral Load 

#### 2.3.1. RNA Extraction and cDNA Synthesis

Total RNA was extracted from whole fish as previously described [30]. Briefly, individual fish were weighed, and 4 mL g^−1^ of fish of a guanidinium thiocyanate-based denaturation solution was added to each fish individually stored in a Whirl-Pak^®^ bag (Nasco, Madison, WI, USA). Each fish was homogenized using a Stomacher^®^ 80 Biomaster (Seward, Ltd., West Sussex, United Kingdom), after which the homogenate was centrifuged for 30 min (2200 rpm). RNA was extracted from 1 mL of the homogenate with phenol-chloroform, precipitated, and resuspended in 50 µL of RNase-free water. RNA was assessed for quality and concentration by spectrophotometry, yielding a mean ± SE total RNA concentration of 671.4 ± 39.4 ng µL^−1^ (range 23.2–2475.5 ng µL^−1^). Complementary DNA (cDNA) was synthesized using M-MLV reverse transcriptase with random hexamer primers [30]. A standard volume of 11 µL of RNA was used in each cDNA reaction, and the final 20 µL of cDNA was diluted 1:5 by adding 80 µL of Rnase-free water. Newly synthesized cDNA was stored at −80 °C.

#### 2.3.2. Viral RNA Quantification via Reverse Transcriptase Real-Time PCR

Viral RNA was quantified using the universal IHNV N gene reverse transcriptase real-time PCR (RT-rPCR) assay as previously described [31]. Briefly, 5 µL of each diluted cDNA sample was combined with forward and reverse primers, TaqMan^®^ FAM-labeled probe for the IHNV N gene, VIC^®^-labeled probe for the artificial positive control (APC) and amplified on an Applied Biosystems ViiA7™ real-time PCR machine (Foster City, CA, USA). APC plasmid DNA was linearized and used to construct a standard curve (ten-fold dilutions from 5 × 10^7^ to 5 DNA copies per RT-rPCR reaction) to quantify the absolute copy number of viral RNA [31]. Each sample was run in duplicate wells and interpreted as positive only when amplification was detected in both replicates within 40 cycles. The analytical sensitivity of the IHNV RT-rPCR assay was determined based on the PCR efficiencies observed for the APC plasmid DNA standard curves for all assays included in this analysis (*y* = m*x* + b). The reaction efficiencies varied from −3.4 to −3.5 (slope, m), and the y-intercept values varied from 39.8 to 40.9 (b), indicating similar limits of detection across the IHNV RT-rPCR assays. The detection limit of the IHNV N gene RT-rPCR assay was 2512.9 viral RNA copies per gram of fish tissue (3.40 log_10_ RNA copies g^−1^). This assay detects both genomic and messenger RNA, and this combined quantity will be referred to hereafter as viral load.

#### 2.3.3. Statistical Analyses

##### Survival of Experimental Host Populations

To assess the influence of viral treatment on host survival, Kaplan–Meier survival analysis with a Mantel-Cox Log-rank test was performed using SigmaPlot v. 13 (Systat Software Inc., San Jose, CA, USA), as previously described [3]. Briefly, survival curves were initially constructed for each set of 20 Chinook salmon fry exposed to a virus or virus-free treatment. Comparison of survival curves between experimental replicates demonstrated no significant differences, with cumulative percent survival for individual replicate tanks reported in Appendix A. Therefore, data were pooled from triplicate groups of 20 fish per treatment to construct a single survival curve for comparison between virus treatments within and across host populations. To identify statistically significant differences between survival curves, multiple pairwise comparisons using the Holm-Sidak test (α = 0.05) were performed in SigmaPlot v. 13. Final graphics were generated using GraphPad Prism v. 8.0 (GraphPad Software Inc., La Jolla, CA, USA).

##### Infection of Experimental Host Populations

To characterize differences in host susceptibility to infection with L, UC, and MD genogroup viruses, binary infection data and viral load data were first analyzed using descriptive statistics. For each host population and virus strain, infection prevalence at 3 dpe is reported as an indicator of virus infectivity. For virus positive fish, the quantities of IHN virus detected at 3 dpe are reported as the mean log_10_ viral load (log_10_ RNA copies g^−1^), indicating early virus replication capability once inside the host.

Exposure of juvenile Chinook salmon to a range of virus concentrations allowed estimation of the dosages needed to infect 50 percent of a host population (ID_50_) with each IHNV strain. As previously described in Breyta et al. [28] and McKenney et al. [32], a logistic regression model was fit to exposure dose to estimate the level at which 50 percent of a host population is expected to become infected. Briefly, these regression analyses were conducted in the statistical program R, v. 3.4.4 [33] using the glm function of the *Stats* package. For ten of the twelve models (three virus strains in four host populations), the binomial error family was used for the logistic regression. For the remaining two models, Up-Spring and Low-Spring Chinook salmon populations exposed to the L virus, a quasibinomial error family was favored because of apparent overdispersion [34]. Estimates of ID_50_ were determined using the dose.p function of the *Mass* package in R [35], and were used as a metric of virus infectivity. Significant differences between ID_50_ values between fish populations and virus strains were assessed using the Welch-Satterthwaite 2-tailed *t*-test [28]. To account for multiple pairwise comparisons, a Bonferroni correction of the 0.05 significance level was performed by dividing it by the number of comparisons performed in a given analysis. Significant differences between ID_50_ estimates were evaluated between viral strains within host populations (three comparisons, Bonferroni α = 0.0167), or between host populations within viral strains (six comparisons, Bonferroni α = 0.0083).

##### The Influence of Exposure Dose, Viral Strain, and Host Population on Infection Frequency

While comparisons of ID_50_ estimates have precedence in evaluating differences in the infectivity of IHNV strains [28,32], a fixed effects logistic regression approach was also taken to evaluate the influence of exposure dose, viral strain, and host population on the frequency of IHNV infection. The inclusion of categorical variables, such as host population in regression models, requires them to be dummy coded first, in which, an indicator variable is created for all but one level of the categorical variable. The level that is held out serves as the reference level. The influence of this reference level on the regression outcome is represented by the model’s y-intercept, whereas the influence of each remaining level is captured by a regression coefficient representing the offset for that level from the y-intercept. Logistic regression models the log odds ratio as a linear effect of the model’s independent variables, which in the present case is formalized as follows:(1)$$\ln\left(OR_{i}\right)=\beta_{0}+\log_{10}\left(dose_{i}\right)\times \beta_{1}+\overset{H}{\underset{h=2}{\sum }}host_{h,i}\times \alpha_{h}+\overset{S}{\underset{s=2}{\sum }}strain_{s,i}\times \gamma_{s}\;$$
where:
$OR_{i}$ is the odds ratio associated with the *i*th fish, the expected number of infected individuals per uninfected individual;$\beta_{0}$ is the model’s y-intercept, capturing the reference level’s influence on the infection outcome;$\log_{10}\left(dose_{i}\right)$ is the log_10_ of the exposure dose for the *i*th fish;$\beta_{1}$ is the effect size of exposure dose on the infection outcome;$host_{h,i}$ is the dummy variable for the *h*th host population for the *i*th fish;$\alpha_{h}$ is the offset from the y-intercept associated with the *h*th host population;$strain_{s,i}$ is the dummy variable for the *s*th viral strain for the *i*th fish;$\gamma_{s}$ is the offset from the y-intercept associated with the *s*th viral strain.

This full regression model evaluated infection status at 3 dpe as the response variable, relative to exposure dose, viral strain, and host population, as explanatory variables. Reduced models omitted either viral strain, host population, or both. Model comparison and selection were based on the minimization of Akaike’s information criterion (AIC), which decreases as the model’s out-of-sample predictive accuracy improves (Appendix A). To identify significant differences between all pairs of levels for both host population and viral strain variables, refitting the same model multiple times was necessary to reassign the reference level. Because multiple pairs were evaluated, Bonferroni correction was once again applied.

#### 2.3.4. The Influence of Exposure Dose, Viral Strain, and Host Population on Viral Load

The influence of exposure dose, viral strain, and host population on viral load was evaluated using a generalized linear model (GLM) in R. The form of the full model closely resembles Equation (1), with the exception that the dependent variable was $\log_{10}\left(viral\;load_{i}\right)$. Reduced models once again omitted either viral strain, host population, or both. Model comparison and selection were also based on the minimization of the AIC (Appendix A). As before, refitting the same model multiple times to reassign the reference level of both categorical variables was necessary to identify significant differences between all pairs of levels.

### 2.4. Analysis of IHNV Field Surveillance Data

Surveillance records were reviewed to characterize the field occurrence patterns of UC and MD IHNV infection in spring-, summer- and fall-run CRB Chinook salmon [4], based on the IHNV Virology, Genotyping, and Surveillance (VGS) database. The VGS database consists of 6766 records representing 1146 sample sites in the CRB and coastal Washington and Oregon and includes 15 different fish hosts. It comprises surveillance records collected between the years 2000–2012 by all five resource management agencies that operate conservation hatcheries across the US Pacific Northwest [4]. Additionally, the VGS database includes viral genotype information for IHN virus isolates, based on the standard IHNV genotyping window of 303 bp mid-G-region. To query the VGS data set for records of Chinook salmon of the CRB, this data set was imported into ArcMap, v. 10.6.1 (Esri, Redlands, CA, USA), and a spatially bounded subset of the database was extracted, focusing on the Columbia River Watershed Boundary [36] using the Clip command. Further subsetting of these data by host species resulted in a total of 1422 records for CRB Chinook salmon, representing 128 total sample sites. Migratory run data were available for 1011 records, which constitute the subset of the VGS database analyzed in this paper. The virus was noted in 25.7% of records, and 59.2% of these had viral genotype data available. Together, these data were used to analyze the prevalence of UC and MD IHNV infection in spring-, summer- and fall-run Chinook salmon populations of the CRB.

### 2.5. Abundance of Spring-, Summer- and Fall-Run Chinook Salmon across the CRB

The relative abundances of spring-, summer- and fall-run Chinook salmon across the CRB were estimated based on the average number of adult fish migrating past the Bonneville Dam between the years 2000–2012 [18]. Host abundance was integrated with IHNV field surveillance data to characterize possible intra-specific patterns of UC and MD IHNV infection across the diverse Chinook salmon life-history forms.

## 3. Results

### 3.1. Survival of Experimental Populations of Columbia River Basin Chinook Salmon

For each of the four Chinook salmon populations included in this investigation, the kinetics of daily cumulative percent survival (CPS) after exposure to L, UC, or MD virus strains are illustrated in Figure 2. When exposed to the positive control L genogroup virus at a high concentration of 2 × 10^5^ PFU mL^−1^, the upper CRB spring-run (Up-Spring) population had the lowest 30-day survival (Figure 2a) with an average CPS of 55%. The upper and lower CRB fall-run (Up-Fall and Low-Fall) populations had similar average CPS of 57% and 60%, respectively. In contrast, the lower CRB spring-run (Low-Spring) population had the highest survival (Figure 2c) with an average CPS of 83%. Statistically significant differences in survival were observed in each Chinook salmon population exposed to the L genogroup virus at a high concentration when compared to those salmon exposed to no virus (*p* < 0.001) (Figure 2). Susceptibility to mortality did not significantly differ between three of the host populations following exposure to a high concentration of the L strain, while the Low-Spring population was statistically less susceptible than each of the other three populations (*p* < 0.001) (Figure 2c).

Cumulative percent survival was higher in all host populations exposed to the L genogroup virus at a moderate rather than a high concentration (Figure 2). Among the four Chinook salmon populations, Up-Fall fish had the lowest survival (Figure 2f), followed by Low-Fall fish (Figure 2h) with CPS averages of 75% and 82%, respectively. Higher survival was observed in the Up-Spring and Low-Spring populations with CPS averages of 97% and 93%, respectively. Survival curve comparisons between each spring-run population and mock treatment groups were not significantly different. Together, exposure to a moderate concentration of the L virus showed that both the Up-Spring and Low-Spring populations were significantly less susceptible to mortality than the Up-Fall population (*p* < 0.001). Comparing all results for L virus exposures, the only consistent differences observed among host populations involved the Low-Spring population, which was significantly less susceptible to mortality than all other populations at the high exposure dose, or to one other population at the moderate dose. The virus was recovered from 5/5 Low-Spring and 11/11 High-Spring fish that died after exposure to L virus, indicating that the lower susceptibility to mortality in Low-Spring fish was not due to lack of infection.

Across all experimental host populations exposed to moderate or high concentrations of the UC virus, the average CPS ranged from 92% to 98%. Similarly, for all host populations exposed to MD virus at moderate or high concentrations, the average CPS ranged from 92% to 100%. Multiple pairwise comparisons of survival curves within and across Chinook salmon populations showed no significant differences in survival following exposure with the UC and MD viral treatments, and neither differed from the virus negative control treatments (Figure 2). After high dose exposures, survival was significantly higher (*p* < 0.001) in all host populations exposed to UC and MD strains of IHNV relative to the L virus. This was also observed for the Up-Fall and Low-Fall populations at the moderate dose. Taken together, the positive control L virus strain was significantly higher in virulence than the UC and MD virus strains. Between the UC and MD virus strains tested, virulence was low and not observed to differ within or across host populations.

### 3.2. Infection of Experimental Populations of Columbia River Basin Chinook Salmon

Infection prevalence and viral load (log_10_ viral RNA copies g^−1^ of fish) were used to evaluate differences in the infectivity and early in-host replication, respectively, for the three strains of IHNV. Three days following exposure, the proportion of fish from each experimental host population infected with each of the virus strains was determined (Figure 3). Exposure dosages ranged from 2 × 10^2^ PFU mL^−1^ to 2 × 10^5^ PFU mL^−1^, for which an increasing dose-response was consistently observed. Of the four Chinook salmon populations, the Up-Spring population had the highest number of virus positive fish overall, totaling 48 individuals (Table 2a). The Low-Spring population had the lowest number of virus positive fish, totaling 30 individuals. Between the Up-Fall and Low-Fall populations, equivalent numbers of virus positive fish were observed with 38 and 39 total individuals, respectively.

As an indication of the ability to replicate early in infection, mean log_10_ viral loads and standard error of the mean (SEM) are reported for virus positive fish within each viral treatment at 3 dpe (Table 3), with individual fish data illustrated in Figure 4. There was little evidence of a dose-response or of differences in mean log_10_ viral loads among IHNV strains within each host population (Figure 4). Therefore, for comparisons among the four host populations, viral load data were analyzed as the total viral RNA copies detected in each host population across all viral strains and exposure concentrations (Table 2a). On a whole-population basis, greater overall quantities of the virus were detected in Chinook salmon of the lower CRB (8.61 and 8.92 log_10_ viral RNA copies) relative to upper CRB (8.15 and 8.27 log_10_ viral RNA copies) fish.

Of the three viral strains tested, the positive control L virus strain infected more fish than the UC and MD virus strains (Table 2b). However, the highest total quantity of virus replication (viral load) was observed with the UC virus strain (Table 2b). Most notably, the total quantity of UC virus detected was greater than the total quantity of L IHNV detected.

### 3.3. Viral Dosages Needed to Infect 50 Percent of a Host Population (ID_50_)

Following exposure to a range of virus concentrations, infection frequencies both below and above 50 percent were attained in all four experimental host populations of Chinook salmon with each viral strain (Figure 3). Estimates of the viral dosages at which 50 percent of a population would become infected (ID_50_) with each virus type were generated using binary infection outcomes for fish at 3 dpe. The ID_50_ values are reported in log_10_ PFU mL^−1^, where lower ID_50_ estimates represent higher virus infectivity because a lower concentration is sufficient to infect 50 percent of a population. For all virus strains in all host populations, ID_50_ estimates varied within a range of approximately 1.5 logs, from 3.46 to 4.96 log_10_ PFU mL^−1^ (Figure 5). ID_50_ estimates were lower overall in the Up-Spring population (Figure 5a) and higher for the Low-Spring population (Figure 5c). While not statistically significant, these differences suggest a trend of higher susceptibility to IHNV infection in Up-Spring relative to the Low-Spring population. In the fall-run populations, greater variation between ID_50_ estimates for the L, UC, and MD virus types was observed.

ID_50_ estimates were compared within each host population to evaluate differences in the infectivity of L, UC, and MD virus types (Figure 5). This revealed only one statistically significant difference between two ID_50_ values. In Low-Fall fish, the L virus was significantly higher in infectivity (*p* = 0.008) than the UC virus strain (Figure 5d). While not statistically significant, a similar pattern can be observed in Up-Fall fish (Figure 5b), suggesting higher infectivity with the L virus in the fall-run Chinook salmon populations.

For each virus strain, a comparison of the same ID_50_ estimates across host populations is shown in Figure 6, again demonstrating only one statistically significant difference between two ID_50_ values. The MD virus was significantly higher in infectivity in Up-Spring Chinook salmon (*p* = 0.0123) when compared to Low-Spring fish (Figure 6c). Multiple pairwise comparisons showed infectivity of the L, UC, and MD virus types to be statistically indistinguishable across experimental host populations. In summary, with two exceptions, little variation was observed among ID_50_ estimates, either within or across host populations.

### 3.4. The Influence of Exposure Factors on Infection Status and Viral Load

The influence of virus strain, and exposure dose on the infection status and viral load was assessed using logistic regression (parameter estimates in Appendix A) and generalized linear models (parameter estimates in Appendix A), respectively. Each of the two models was refit three times to allow comparisons between all pairs of virus strains (L vs. UC, L vs. MD, and UC vs. MD). To account for multiple pairwise comparisons, the 0.05 significance level was Bonferroni-corrected to α = 0.0166. Differences between all virus strain pairs are reported as *p* values in Table 4. The models indicated a statistically significant overall influence of exposure dose on infection status (*p* < 0.001), but not on viral load (*p* = 0.115) (Table 4). Consistent with these analyses, an increasing dose-response can be observed in the frequency of virus positive fish at 3 dpe (Figure 3). Virus strain was not observed to significantly influence infection status or viral load (Table 4). However, the logistic regression and generalized linear models reported *p* values that were above the Bonferroni-corrected α = 0.0166, but less than 0.05 for comparisons between L and UC virus strains. While statistical power might be insufficient to detect significant differences in the influence of virus strain on infection status and viral load, the *p* values reported by the logistic regression and generalized linear models may suggest biologically relevant differences between L and UC viruses, with L being more infectious (*p* = 0.030) and UC replicating to higher viral load (*p* = 0.042). This is consistent with general observations in Table 2b, where the L virus infected the greatest number of fish and UC the fewest, but the total virus quantities were highest for the UC virus.

### 3.5. The Influence of the Host Population on Infection Status and Viral Load

The influence of the host population on infection status and viral load was also assessed using logistic regression (Appendix A) and generalized linear models (Appendix A), respectively. Here, each of the two models was refit six times to compare all population pairs. To account for multiple pairwise comparisons, the 0.05 significance level was Bonferroni-corrected to α = 0.0083. Differences between all pair levels are reported as *p* values in Table 5. The logistic regression models indicated no statistically significant influence of the host population on infection status for most host pairs, with the exception of one pair contrasting Up-Spring and Low-Spring Chinook salmon (*p* < 0.001) (Table 5). Consistent with this exception, Up-Spring Chinook salmon were observed to have greater numbers of virus positive fish than the Low-Spring population (Table 2a).

The generalized linear models, used to evaluate the influence of the host population on viral load, also detected only one statistically significant difference, again between Up-Spring and Low-Spring Chinook salmon (*p* < 0.005) (Table 5). However, while higher numbers of Up-Spring Chinook salmon were infected with IHNV at 3 dpe (Table 2a), greater overall quantities of the virus were detected in Low-Spring Chinook salmon (Table 2b). Nevertheless, these data suggest that in this one case, the host population has an influence on viral load, with infection of Low-Spring fish leading to significantly higher viral loads than infection of Up-Spring fish. While statistical power may be insufficient to detect other significant differences in viral loads among the host populations, the GLM reported *p* values that were above the Bonferroni-corrected α = 0.0083, but less than 0.05 for three additional pairs of levels (Table 5). Each of these additional population pairs included a fall-run Chinook salmon population from either the upper CRB (*p* = 0.027) or lower CRB (*p* = 0.008, 0.05). Across all three contrasting population pairs, the lower CRB populations were observed to have higher total quantities of the detectable virus at 3 dpe (Table 2a). Together, these data suggest a non-significant trend that geographic distribution of a host population across the lower CRB may have an influence on viral load outcome.

### 3.6. Persistence of IHNV Infection in Juvenile Columbia River Basin Chinook Salmon

To provide a preliminary indication of variation in viral persistence, infection prevalence and mean log_10_ viral loads (log_10_ RNA copies g^−1^ of fish) were determined for a subset of fish that survived exposure to the high concentration of each viral strain. Thirty days following exposure, detectable infections with L, UC, and MD IHNV were observed to persist in a proportion of fish from each experimental host population (Table 6). Although statistical analyses were not conducted, due to the small numbers of survivors analyzed (*n* = 8) for each treatment group, the following trends were observed. Persistent infection prevalence was highest among fish exposed to the positive control L virus, relative to the UC and MD viral treatments. Similarly, mean log_10_ viral loads were generally highest among fish infected with the L IHNV strain. The total quantity of L IHNV detected at 30 dpe in all host populations combined was more than two logs, higher than the total quantities of the UC or MD viruses. While fewer fish were observed to be infected with the UC and MD IHNV strains, the mean log_10_ viral loads of fish infected at 30 dpe with the UC and MD viral strains were not inconsequential. Among Chinook salmon persistently infected with the UC IHNV strain, mean log_10_ viral loads ranged between 4.38–5.83 log_10_ RNA copies g^−1^ of fish. Among fish persistently infected with the MD IHNV strain, mean log_10_ viral loads ranged between 5.18–5.87 log_10_ RNA copies g^−1^ of fish. Using our previously published ratio for detection of viral RNA by RT-qPCR versus detection of infectious virus by plaque assay [3], these values for UC and MD persistence in fish at 30 dpe would be in the range of approximately 1000 plaque-forming units g^−1^ of fish. Overall, higher numbers of virus positive fish were observed with the UC IHNV strain relative to the MD strain. Further, higher total virus quantities were observed with the UC IHNV strain relative to the MD viral strain.

The total numbers of virus positive fish at 30 dpe from each experimental host population ranged between 4–10 fish or 16.7–41.7% of fish tested, respectively (Table 6). Among the four Chinook salmon populations, the Up-Spring population had the highest total number of persistently infected fish (*n* = 10 out of 24), whereas the Low-Spring population had the lowest total number of persistently infected fish (*n* = 4 out of 24) at 30 dpe. Again, statistical analyses were not conducted, due to the small numbers of fish tested, but consistent with infection frequencies reported at 3 dpe, infection persistence was lower in the Low-Spring population than in the contrasting three host populations.

The total quantities of virus detected at 30 dpe among the four host populations ranged between 6.89–9.15 log_10_ total viral RNA copies. Total virus quantities were approximately 1–2 logs higher in the two upper CRB Chinook salmon populations than in the two lower CRB populations. As suggested by generalized linear models, that evaluated the influence of the host population on early viral load, these data support a non-significant observation that the geographic distribution of a host population may influence viral load outcome. In general, viral persistence and infection prevalence at 30 dpe are consistent with other quantitative observations of intraspecies variation in CRB Chinook salmon susceptibility to IHNV infection, as described above.

### 3.7. Field Occurrence Patterns of UC and MD IHNV Infection in CRB Chinook Salmon

Of the three migratory run-types of adult CRB Chinook salmon, fall-run fish are most abundant (60%), followed by spring-run fish (27%), and summer-run fish comprise a minority (13%), based on avg. annual returns for 2000–2012 [17]. The prevalence of IHNV infection in spring-, summer- and fall-run Chinook salmon populations of the CRB was determined as the proportion of virus positive IHNV cohorts relative to the total number of cohorts tested for virus, for each life-history phenotype (Table 7). Across Chinook salmon populations of the CRB, infection prevalence did not appear to differ between the spring- and fall-run phenotypes, with 27% and 25%, respectively. However, infection prevalence was lower in CRB summer-run populations (13%), which also had the fewest number of cohorts tested. Genotyping was conducted for 60% of the virus positive samples from either spring- or fall-run fish. For summer-run fish, only two of the virus positive samples were genotyped (25%).

The specific prevalence of UC and MD IHNV infection across Chinook salmon populations of the CRB was determined as the proportion of UC or MD IHNV positive cohorts relative to the total number of IHNV positive cohorts for which genotype data were available (Table 7). Among all CRB Chinook salmon populations, the prevalence of IHNV infection was higher with UC genogroup viruses (83% overall) than with MD viruses (17% overall). For both spring- and fall-run CRB Chinook salmon, infection prevalence was higher with UC genogroup viruses (82% and 88%, respectively) than with MD viruses (18% and 12%, respectively). In summer-run fish, both UC and MD were detected, each in one sample.

## 4. Discussion

This investigation examined the host-pathogen interactions that drive the ecology of IHNV in the CRB, where Chinook salmon are by far the most abundant virus host. We, therefore, compared four CRB Chinook salmon populations selected to represent the phenotypic and genetic diversity of Chinook salmon in the CRB, using UC and MD subgroup IHNV strains that occur in the CRB. We also included an L group strain as a positive control known to be highly adapted and virulent in Chinook salmon. This differs conceptually from our previous work [3] by focusing on intra-specific variation in IHNV susceptibility of Chinook salmon in the CRB. The experimental design used here also expanded on our previous methods by assessing virulence using two exposure doses and conducting a substantially more robust analysis of infection using four exposure doses, allowing us to more thoroughly quantify the infection. This provided an improved ability to discern fitness differences, if they exist, among host populations and among virus strains.

In our results, disease in the host was measured as mortality after exposure to either high or moderate doses of virus in immersion challenges, and viral infection was assessed by several measures, including the frequency of initial infection, viral loads at 3 days post exposure, and persistence of detectable infection through 30 days. The quantification of multiple traits for three divergent virus strains in four diverse host populations generated an extensive and complex data set. Our analysis of those data was designed in terms of testing two null hypotheses: (1) There is no variation among the four host populations, and (2) there is no variation among the three virus strains. Our results do not support either of these hypotheses, but rather suggest a “general rule” in each case, with some exceptions.

From the host perspective, the majority of tests described here demonstrate no significant difference between the four populations tested. For example, measurements of mortality among the four host populations provided 36 possible comparisons where a pair of hosts could differ (six possible host pairs, three virus strains, two challenge doses), and only five significant differences were found, all for L virus. Similarly, analyses of ID_50_ values in the four host populations provided 18 possible comparisons where a pair of hosts could differ (6 possible host pairs, three virus strains), and only one of the 18 comparisons found a significant difference. To summarize what might be learned from the exceptions, specific cases where variation was observed among host populations are shown in Table 8a, presented as significant differences, trends, or non-statistical quantitative observations. For survival measures, the Low-Spring population was less susceptible than other populations to mortality after L virus exposure (Table 8a). Similarly, several infection measures indicated that the Low-Spring population was less susceptible to infection than other populations. However, paradoxically, the Low-Spring population had higher viral loads at 3 days post infection than other host populations. Some measures indicated that the Up-Spring population was more susceptible to infection than other populations, most commonly the Low-Spring, but this was not reflected in lower survival. Thus, Table 8a shows some consistent indications for lower susceptibility of the Low-Spring to both infection and mortality, and possibly higher susceptibility to infection for the Up-Spring population. However, it is important to remember that even for these populations the majority of comparisons did not show differences.

Thus, while our data do not completely support the null hypothesis of no variation in IHNV susceptibility among host populations along every dimension, there is very little significant variation between them overall. Based on the finding that the Low-Spring population is somewhat less susceptible to both mortality and infection, while the Up-Spring is more susceptible, and both Fall populations were intermediate, we did not find any generalized correlation of IHNV susceptibility with the spring- or fall-run life-history phenotypes. Thus, we do not confirm the trend suggested in Hernandez et al. [3], where the single spring-run population (stream-type juveniles) generally seemed more susceptible to infection than the fall-run (ocean-type) population. Similarly, we do not observe any consistent differences associated with the geographic origin of the host populations in the upper or lower CRB. Rather, the variation observed appears to be unique to individual host populations, driven by factors other than life-history phenotype or geographic distribution.

From the virus perspective, there were some clear differences in virulence measures, specifically for the L virus, but the great majority of infectivity measures did not show significant differences among the three virus strains. For example, ID_50_ values for each virus strain provided 12 possible comparisons where two virus strains might differ (three possible pairs of virus strains, compared to each of four hosts), and only one comparison had a significant difference. Table 8b summarizes the exceptions observed from the virus perspective. In general, there is a consistent pattern of significant differences and trends indicating higher virulence and higher infectivity for the L virus relative to UC and MD. However, differences in infectivity were only significant for specific virus pairs compared to some host populations. Again, the early viral load measure did not correlate with higher infection frequency, with a non-significant trend of higher viral load for UC virus relative to L at 3 days post exposure. However, by 30 days the L virus had both the highest infection frequency and highest viral loads. Most important for understanding virus ecology in the CRB, there were no differences observed between the UC and MD viruses in any measure tested here, for either virulence or infectivity. Thus, our data confirm the null hypothesis of no variation in virulence or infectivity for the UC and MD virus strains, but by several measures, the positive control L virus differs from the other two virus strains.

For the experiments described here, we used a single virus isolate to represent each of three IHNV genetic subgroups. The isolates used were chosen from major IHNV epidemic events and are the index isolates of the most dominant IHNV genotypes circulating broadly in Chinook salmon of the CRB of Washington, Idaho, and Oregon (mG001U for UC, mG110M for MD), and the Sacramento River watershed in California (mG011L for L). Genetic surveillance of 771 IHNV isolates from CRB Chinook salmon has found 199 viruses with genotype mG001U collected between 1985–2010, and 27 viruses with genotype mG110M collected between 2007–2013 (Molecular Epidemiology of Aquatic Pathogens IHNV database at http://gis.nacse.org, accessed on 8 March 2019) [37]. Similarly, genotyping of 165 IHNV isolates from Chinook salmon in California and Southern Oregon has found genotype mG011L 59 times between 1986–2004. Therefore, these represent the most relevant virus genotypes for examining the host-pathogen interactions of CRB Chinook salmon with UC, MD, and L IHNV. However, any individual virus isolate may not be representative of a genotype or genetic subgroup, so our broader interpretations of the data involve an assumption that these specific isolates are representatives. Confirmation of this would require additional testing of specific findings with more virus isolates in the future.

The experimental studies presented here involve some caveats that may impact the interpretation of the results. Although the virus isolates used for the work were selected as optimal representatives of the three major IHNV genogroups in North America (U, M, and L), they were not obtained directly from Chinook salmon in the CRB (Table 1b). Therefore, additional studies using UC and MD virus isolates obtained more recently from CRB Chinook salmon would be of interest, and are currently underway. Additionally, we used eight fish per exposure dose for the infection study, and a larger number may have enhanced our ability to detect subtle differences in infectivity. However, the use of four exposure doses provided results that bracketed 50% infection for all virus and host combinations, allowing us to calculate ID_50_ values without extrapolation beyond the results, as is sometimes required in similar studies. Finally, the determination of infection status at a single sampling time point 3 dpe generated a “snap-shot” of relative infection levels that may not capture all relevant differences if virus isolates vary in their speed of replication in different Chinook salmon hosts. The choice of 3 dpe as an optimal first sampling time point to test was based on previously published time course studies of U and M virus replication in both rainbow trout [38] and sockeye salmon [39]. In those experiments, the sampling point at 3 dpe provided a balance between allowing both virus strains to reach peak in-host titers, but avoiding substantial impacts of viral clearance that are evident by day 7 for some strains. We do not know if the 3 dpe time point is optimal for IHNV in Chinook salmon, so more complex time course studies with multiple time points will be useful and are currently underway.

The data from controlled laboratory infection experiments are interesting in comparison to the field occurrence patterns presented in Table 7. Field surveillance records for IHNV across the CRB showed that IHNV prevalence was comparable at 25–27% for the two dominant Chinook salmon life-history types (spring-run and fall-run) present across the watershed. This fits with our experimental data, where we found little variation in infection measures overall, and no differences that correlated with the spring- or fall-run life-history types among the four Chinook salmon populations that were tested. However, assessment of field occurrence patterns specific for the UC and MD subgroups found a much higher prevalence of UC in CRB Chinook salmon relative to MD IHNV. Perhaps the most interesting observation from our study is that this strong field pattern was not reflected in the results of our experimental studies. In the field, IHNV in spring-run and fall-run Chinook salmon was 82–88% UC, and only 12–18% MD (Table 7), and yet we found no statistically significant differences between UC and MD viruses in any host population, by any experimental measure of either virulence or infectivity. The only potentially relevant differences were non-statistical observations that UC infection resulted in higher viral loads than either L or MD infection (Table 2), and UC infection persisted in more fish than MD infection at 30 days post infection (Table 6).

While virus strain infectivity quantified in these controlled laboratory exposures does not explain the disproportionately higher prevalence of UC versus MD IHNV detected in surveillance of CRB Chinook salmon, in-host replication may instead influence field occurrence patterns. The total number of fish infected with UC virus was lower than observed for the MD and L viruses at 3 dpe, but those fish infected with the UC virus constituted the highest total viral quantities observed across the three virus types (Table 2). Thus, UC virus may replicate to higher levels than MD and L IHN virus in these hosts. Together, the abundance of Chinook salmon across the CRB and potentially higher capacity to replicate and shed UC virus may be linked to the higher prevalence of UC infections observed in Chinook salmon populations across the Columbia River watershed. In experimental persistence at 30 days after challenge (Table 6), UC infected more fish than MD, but with only a slightly higher viral load. The reduced viral loads at day 30 compared with day 3 post exposure (compare Table 3 and Table 6) suggest two possible processes; clearance of viral infections and/or the shedding of the virus. Current studies are ongoing to understand how in-host replication, viral shedding kinetics, and persistence may differ between UC and MD viruses in CRB Chinook salmon.

Although there was little intra-specific variation in host susceptibility observed here, it is interesting to consider possible explanations for the exceptions observed as lower susceptibility of the Low-Spring population to both infection and disease, and higher susceptibility of the Up-Spring population to infection. One explanatory factor to consider is genetic differences in the four host populations and the phylogenetic lineages that they represent. Phylogenies of large numbers of CRB Chinook salmon populations are consistent in finding that upper CRB Spring-run Chinook salmon populations form a monophyletic lineage that is genetically highly divergent from all other lineages [7,9]. This may be relevant to the indications of higher susceptibility of the Up-Spring population relative to populations from the other three lineages. In addition, the Up-Spring population is federally listed as endangered under the Endangered Species Act (ESA), and has been reported to have measurably lower genetic diversity relative to other CRB Chinook salmon populations [7,9]. Because innate and adaptive antiviral immune response systems in teleost fishes are controlled by highly polymorphic genetic systems [40,41], higher susceptibility to IHNV infection of the Up-Spring population may be linked to the lower genetic diversity reported for this and other populations of upper CRB spring-run fish that have experienced severe (or repeated) bottlenecks in populations size [42].

Another factor that may be relevant to the variations observed in host susceptibility is differential historical exposure of the four Chinook salmon populations to IHNV. Although the specific experimental fish groups tested here were all hatched from disinfected eggs and reared under controlled conditions to ensure they were IHNV-free, the four Chinook salmon populations they represent all originated from fish hatcheries that have had historical exposure to IHNV at different levels. These hatcheries are all located in sub-regions of the CRB that have frequent UC virus detections, but only the lower CRB populations have had frequent MD virus exposure [15]. Based on IHNV surveillance and genotyping records from throughout the CRB during the years 2000–2012 [4], the Low-Spring population from the Willamette River sub-basin has had a higher frequency of historic exposure to MD virus, carried by steelhead trout, than the Low-Fall populations from the Cowlitz River sub-basin [15], as discussed with R. Breyta via personal communication. Although speculative, it is possible that greater historic exposure of the Low-Spring population to MD IHNV has resulted in selection for greater resistance that is cross-protective against L and UC viruses, observed here as less susceptibility to infection and disease. This might also result in the selection of greater tolerance to IHNV, which may explain the higher viral loads observed in the Low-Spring population.

As potential broader impacts of the findings reported here, the Chinook salmon populations tested in these experiments represent four of the seven distinct Evolutionarily Significant Units designated for CRB Chinook salmon management and conservation [5]. Three of these populations are listed as threatened, and one is listed as endangered under the ESA [9]. Although they were not susceptible to significant disease caused by UC and MD virus strains, they were all susceptible to infection, and the evolution of viral variants with increased virulence is a possibility that should be monitored. In addition, our results are relevant to the possible future scenario of an L genogroup IHNV invasion of the CRB, which could occur if ocean distributions patterns of California Chinook salmon shift northward in response to a reduction in favorable thermal habitats at sea as a result of climate change [43,44]. Based on our results, it is highly likely that all CRB Chinook salmon populations are susceptible to mortality caused by L genogroup IHNV, and all would suffer increased disease impacts if L viruses invaded the CRB. However, the observation of some intra-specific variation in susceptibility to the L virus strain suggests that Spring-run populations may vary in their vulnerability. Thus, although Spring-run Chinook salmon in the CRB are generally less numerous and more imperiled than fall-run fish, some spring-run populations, such as the Low-Spring population tested here, may have an advantage over other populations, with lower disease impacts in the event of an L virus invasion. In general, a sound understanding of both the inter-specific and intra-specific variation in IHNV susceptibility of Pacific salmonid host species will benefit the management of IHN disease in Western North America.

In summary, this investigation comprehensively examined intraspecific variation in CRB Chinook salmon susceptibility to infection and mortality following exposure to U, M, and L strains of IHNV. Results of this investigation generally supported our null hypothesis of no variation among host populations, but we did identify some intraspecific variation in CRB Chinook salmon susceptibility to infection in specific cases. Because infectivity assays demonstrated few statistically significant differences between the UC, MD, and L IHNV, we conclude that the low virulence of UC and MD IHNV in CRB Chinook salmon is not driven by the inability of these viruses to enter juvenile fish, but rather the ability of juvenile fish to control viral infections. When considering differences in UC and MD IHNV prevalence in Chinook salmon across the Columbia River watershed, we conclude that virus infectivity does not appear to be driving the asymmetric field occurrence patterns reported for Chinook salmon. Other factors that were not measured here, such as longer-term in-host virus persistence or viral shedding, may be influencing the large-scale epidemiological patterns observed for IHNV in the CRB. Additional studies are ongoing to better understand the host-pathogen dynamics of CRB Chinook salmon and UC and MD IHNV.

## Figures and Tables

**Figure 1 viruses-13-00701-f001:**
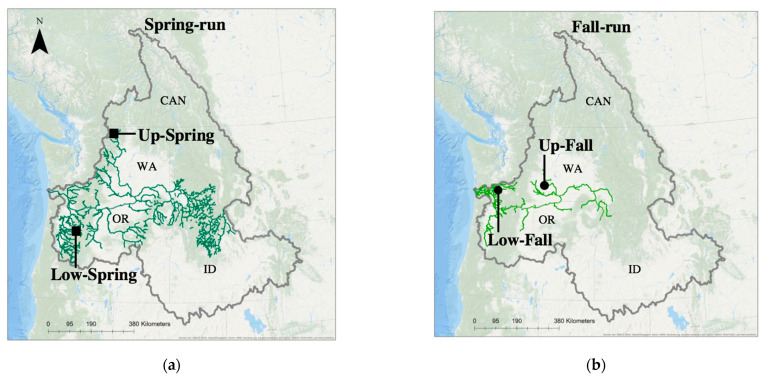
Maps of the Columbia River Basin (outlined in grey) of the northwestern United States and southwestern Canada showing rivers and streams (green lines) containing spring-run (**a**) and fall-run (**b**) populations of Chinook Salmon. Spring-run Chinook salmon migrate to the upper, interior reaches of the basin and spawn in smaller tributary streams, whereas fall-run salmon spawn in larger, mainstem rivers. The lower Columbia Basin (“Low”) lies west of the Cascade Mountain Range, and the upper Columbia Basin (“Up”) lies east of the Cascade Mountains. Spatial distributions of the Chinook salmon populations were sourced from StreamNet GIS Data [10]. Source locations for the four Chinook salmon populations tested here are shown.

**Figure 2 viruses-13-00701-f002:**
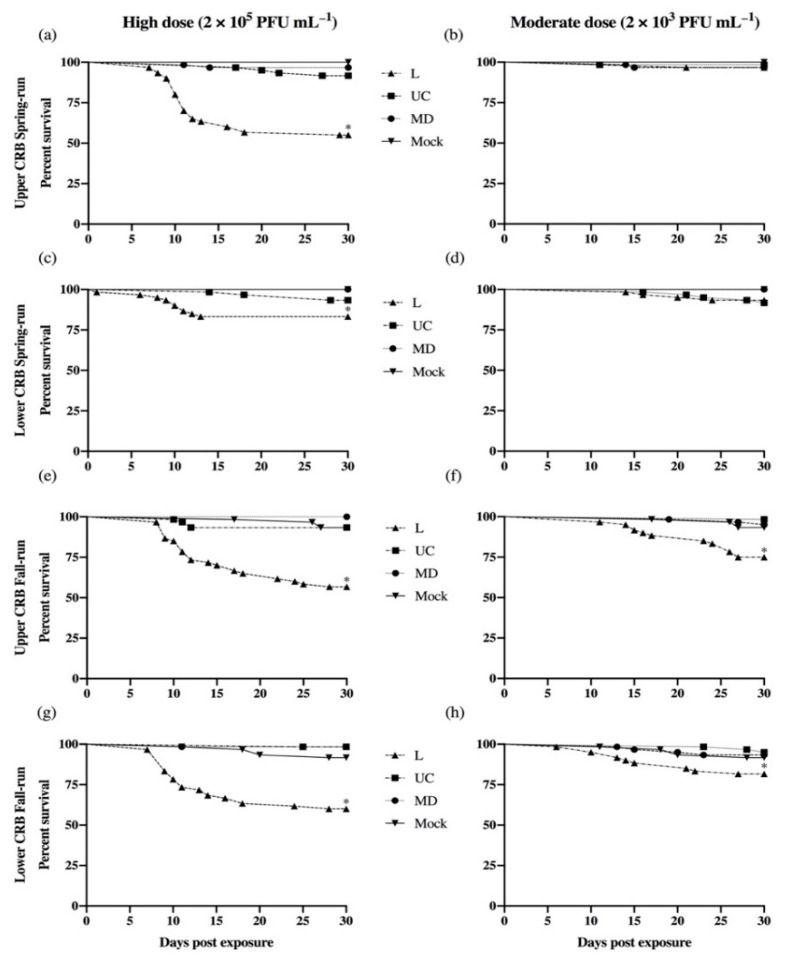
Daily cumulative percent survival of juvenile spring-run (**a**–**d**) and fall-run (**e**–**h**) Chinook salmon (*Oncorhynchus tshawytscha*) of the upper and lower Columbia River Basin (CRB) exposed to L, UC, and MD strains of infectious hematopoietic necrosis virus (IHNV) or virus-free media (Mock). Experimental host populations were exposed to each virus strain at high (left) and moderate (right) doses by static immersion in 1 L. Survival curves were constructed by pooling data from triplicate groups of 20 fish per treatment. Asterisks indicate that fish in the positive control L genogroup (FR0031) IHNV treatment had significantly lower survival than fish in the other viral strain treatments (*p* < 0.001). No significant differences were observed between the MD and UC virus treatments within or across host populations.

**Figure 3 viruses-13-00701-f003:**
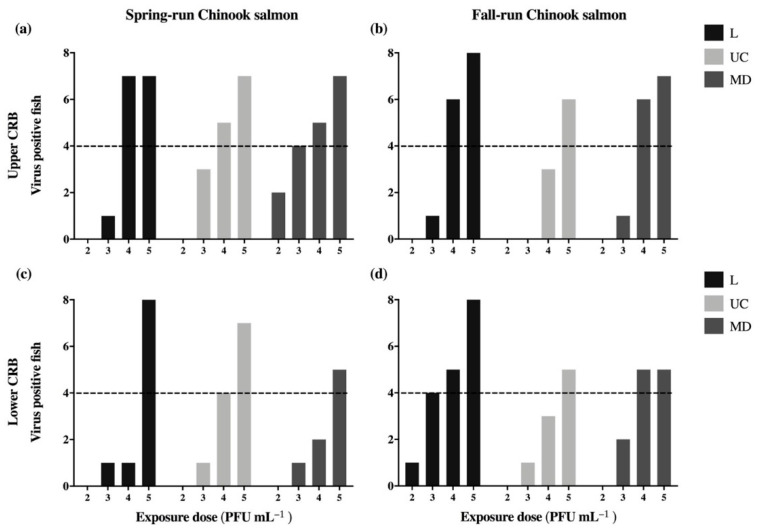
Virus positive spring-(**a**,**c**) and fall-run (**b**,**d**) Chinook salmon of the upper (**a**,**b**) and lower (**c**,**d**) Columbia River Basin at 3 days post exposure to L, UC, and MD strains of IHNV by static immersion. Exposure dosages of 2 × 10^2^ PFU mL^−1^ through 2 × 10^5^ PFU mL^−1^ are reported along *x*-axes as exponents of viral challenge concentrations. The *y*-axis indicates the number of virus positive fish out of a total of 8 fish per treatment group, and dotted lines indicate 50% infection.

**Figure 4 viruses-13-00701-f004:**
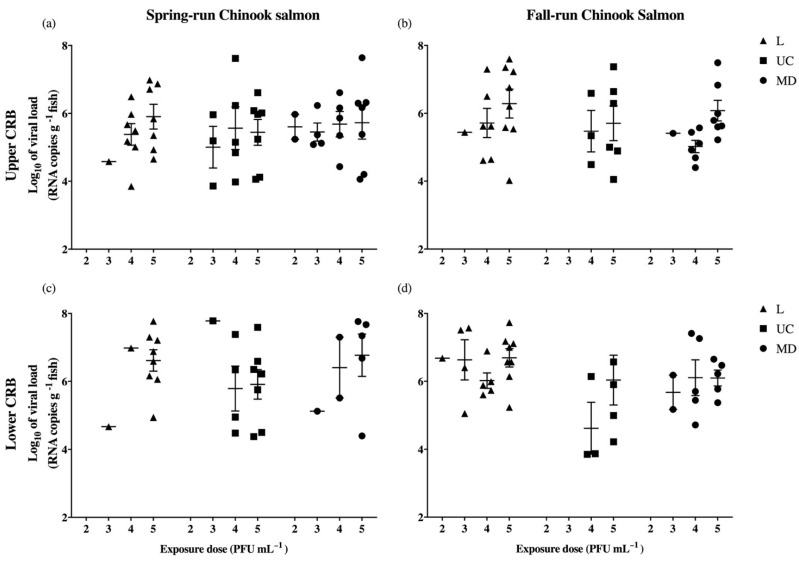
Viral load (log_10_ RNA copies g^−1^ of fish) of IHNV positive spring-(**a**,**c**) and fall-run (**b**,**d**) Chinook salmon of the upper (**a**,**b**) and lower (**c**,**d**) Columbia River Basin at 3 days post exposure (dpe) to L, UC, or MD strains of IHNV. Exposure dosages are reported along *x*-axes as exponents of viral challenge concentrations (2 × 10^2^–2 × 10^5^ PFU mL^−1^). Mean log_10_ viral loads (±SEM) are indicated for each group of virus positive fish. Quantities of IHN virus detected at 3 dpe are reported in Table 3.

**Figure 5 viruses-13-00701-f005:**
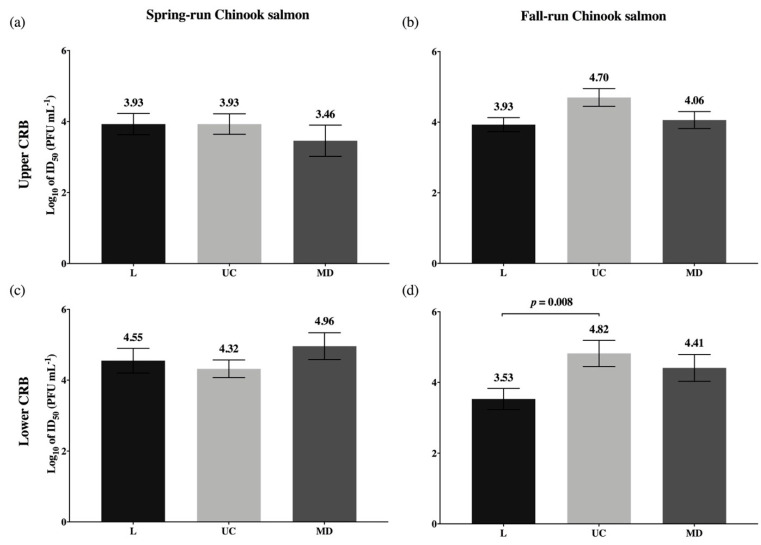
Estimates of viral dosage needed to infect 50 percent (ID_50_) of juvenile Chinook salmon with L, UC, and MD strains of IHNV by batch immersion. ID_50_ estimates (log_10_ PFU mL^−1^, ±SE) are reported for spring-(**a**,**c**) and fall-run (**b**,**d**) Chinook salmon of the upper (**a**,**b**) and lower (**c**,**d**) Columbia River Basin at 3 days post exposure. Significant differences between the ID_50_ values generated are indicated as Welch-Satterthwaite 2-tailed *t*-test *p* values. To account for multiple pairwise comparisons, the 0.05 significance level was Bonferroni-corrected to α = 0.0166.

**Figure 6 viruses-13-00701-f006:**
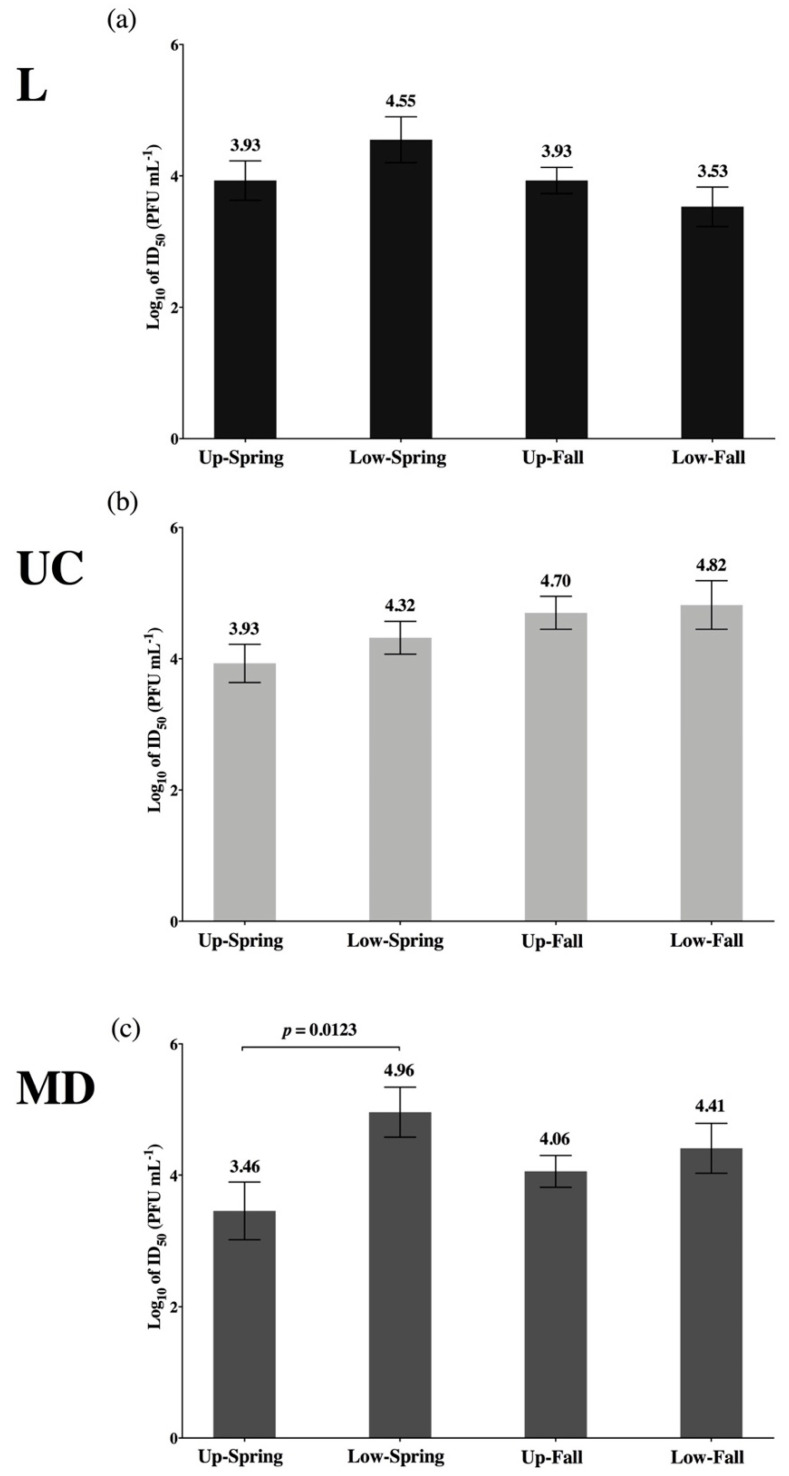
Comparison of ID_50_ values generated for the L (FR0031), UC (RB1), and MD (QTS07) strains of IHNV included in this investigation. ID_50_ values for the L-(**a**), UC-(**b**), and MD-IHNV (**c**) strains were assessed for differences in infectivity relative to the experimental host population. Multiple pairwise comparisons of ID_50_ (log_10_ PFU mL^−1^, ±SE) for spring-run (Spring) and fall-run (Fall) *O. tshawytscha* of the upper (Up) and lower (Low) Columbia River were performed. Significant differences between ID_50_ values are reported as Welch–Satterthwaite 2-tailed *t*-test *p* values. To account for multiple pairwise comparisons, the 0.05 significance level was Bonferroni-corrected to α = 0.0083.

**Table viruses-13-00701-t001-a:** 

(a)				
Columbia River Basin Geographic Region ^a^	Adult Migration Timing	Juvenile Life-History	Phylogenetic Lineage ^b,c^	Chinook Salmon Population ^c^
Upper CRB	Spring-run	Stream-type	Interior Columbia River Spring (3)	Methow River (28)
Upper CRB	Fall-run	Ocean-type	Interior Columbia River Fall (2)	Hanford Reach (14)
Lower CRB	Spring-run	Stream-type	Willamette River Spring (1)	North Santiam River (8)
Lower CRB	Fall-run	Ocean-type	Lower Columbia River Fall (1)	Cowlitz River (1)

**Table viruses-13-00701-t001-b:** 

(b)					
Virus Strain	Genogroup, Subgroup	midG Type ^d^	Host	Isolation Site	Year of Isolation
FR0031	L	mG011L	Chinook salmon	Feather River Hatchery, CA	2000
QTS07	MD	mG110M	Steelhead trout	Salmon River Hatchery, WA	2007
RB1	UC	mG001U	Steelhead trout	Round Butte Hatchery, OR	1975

^a^ Populations sourced east of the Cascade Mountain Range represent upper CRB fish, and those obtained west of the Cascade Range constitute lower CRB fish. ^b^ Four major phylogenetic lineages for Chinook salmon populations of the CRB, as in Waples et al. [7]. Parentheses show corresponding CRB Chinook salmon genetic lineage number, as in Narum et al. [9]. ^c^ Chinook salmon populations were selected based on Narum et al. [9], with specific population numbers in parentheses. ^d^ Genotyping data based on 303 nt of the IHNV glycoprotein gene, referred to as the variable midG region [1].

**Table viruses-13-00701-t002-a:** 

(a)		
Host Population	Total # Fish Positive/Total # Exposed, for All Virus Strains	Log_10_ of Total Virus (Viral RNA Copies)
Up-Spring	48/96	8.15
Up-Fall	38/96	8.27
Low-Spring	30/96	8.61
Low-Fall	39/96	8.92

**Table viruses-13-00701-t002-b:** 

(b)		
Virus Strain	Total # Fish Positive/Total # Exposed, for All Host Populations	Log_10_ of Total Virus (Viral RNA Copies)
L	58/128	8.65
UC	45/128	8.91
MD	52/128	8.49
mock	0/30	na

**Table 3 viruses-13-00701-t003:** Quantities of IHN virus detected in 1 g spring- and fall-run Chinook salmon of the upper and lower CRB at 3 days post exposure to L, UC, or MD strains of IHNV.

		L		UC		MD	
		(FR0031)		(RB1)		(QTS07)	
Host Population	Dose ^a^	Mean Log Viral Load ^b^	SEM	Mean Log Viral Load ^b^	SEM	Mean Log Viral Load ^b^	SEM
Up-Spring	2 × 10^2^	-	-	-	-	5.61	0.365
	2 × 10^3^	4.58	na	5.00	0.613	5.45	0.268
	2 × 10^4^	5.38	0.316	4.56	0.628	5.68	0.374
	2 × 10^5^	5.90	0.364	4.44	0.38	5.72	0.482
	Mock	-	-	-	-	-	-
Up-Fall	2 × 10^2^	-	-	-	-	-	-
	2 × 10^3^	5.44	na	-	-	5.41	na
	2 × 10^4^	5.72	0.429	5.47	0.61	5.02	0.18
	2 × 10^5^	6.29	0.426	5.71	0.513	6.08	0.30
	Mock	-	-	-	-	-	-
Low-Spring	2 × 10^2^	-	-	-	-	-	-
	2 × 10^3^	4.67	na	7.78	na	5.12	na
	2 × 10^4^	6.98	na	5.79	0.662	6.41	0.9
	2 × 10^5^	6.61	0.315	5.91	0.434	6.77	0.6
	Mock	-	-	-	-	-	-
Low-Fall	2 × 10^2^	6.68	na	-	-	-	-
	2 × 10^3^	6.63	0.592	8.43	na	5.68	0.505
	2 × 10^4^	6.02	0.228	4.62	0.76	6.11	0.527
	2 × 10^5^	6.69	0.269	6.04	0.734	6.10	0.234

^a^ Exposure dosages are reported in plaque-forming units (PFU) mL^−1^. ^b^ Mean log_10_ viral load (log_10_ RNA copies g^−1^ of fish) is reported for virus positive fish within each treatment group. A dash is used where no fish were positive. SEM is the standard error of the mean.

**Table 4 viruses-13-00701-t004:** *p* values for regression coefficients examining the relationship between IHNV strains and exposure dose on infection status (below diagonal) and log_10_ viral load (above diagonal), assessed using logistic regression and generalized linear models (GLMs), respectively ^a^.

IHNV Genogroup, Subgroup	L	UC	MD
(Experimental Virus Strain)	(FR0031)	(RB1)	(QTS07)
L			
(FR0031)	-	0.042	0.330
UC			
(RB1)	0.030	-	0.280
MD			
(QTS07)	0.315	0.237	-
Exposure Dose	0.001 * (infection status); 0.115 (log_10_ viral load)		

^a^ Both models were refit three times to allow comparisons between all pairs of strains. To account for multiple pairwise comparisons, the 0.05 significance level was Bonferroni-corrected to α = 0.0166. Differences between all virus strain pairs are reported as *p* values. Virus strain was not observed to influence the frequency of infection or viral load. The logistic regression model indicated a statistically significant influence of exposure dose on the frequency of infection, noted by an asterisk (*p* < 0.001 *). In contrast, exposure dose does not appear to influence viral load.

**Table 5 viruses-13-00701-t005:** *p* Values for regression coefficients examining the relationship between experimental host population and infection status (below diagonal) and log_10_ viral load outcome (above diagonal), assessed using logistic regression and generalized linear models (GLMs), respectively ^a^.

Chinook Salmon Population	Up-Spring	Up-Fall	Low-Spring	Low-Fall
Up–Spring	-	0.576	0.005 *	0.008 ^b,c^
Up–Fall	0.056	-	0.027 ^c^	0.049 ^c^
Low–Spring	0.001 *	0.119	-	0.707
Low–Fall	0.085	0.847	0.080	-

^a^ Each of the two models were refit six times to compare all pairs of levels. To account for multiple pairwise comparisons, the 0.05 significance level was Bonferroni-corrected to α = 0.0083. Differences between all pair levels are reported as *p* values. The experimental host population was observed to statistically influence the frequency of infection (*p* < 0.001) and log_10_ viral load (*p* < 0.005), but only when contrasting upper and lower CRB spring-run Chinook salmon. Asterisks denote significant differences. ^b^ The *p* value shown as 0.008 was actually 0.00849, and thus, not below the Bonferroni-corrected level of significance of 0.0083. ^c^ As noted in the text, three *p* values were not below the Bonferroni-corrected level of significance of 0.0083, but were less than *p* = 0.05.

**Table 6 viruses-13-00701-t006:** Persistence of IHNV infection in juvenile CRB Chinook salmon at 30 days post exposure to L, UC, and MD strains of IHNV at a single high dose of 2 × 10^5^ plaque-forming units mL^−1^.

	L (FR0031)			UC (RB1)			MD (QTS07)				
Host Population	Num.pos. /Tested	Mean Log Viral Load	SEM ^a^	Num.pos. /Tested	Mean Log Viral Load	SEM ^a^	Num.pos./Tested	Mean Log Viral Load	SEM ^a^	Total Fish pos.	Log_10_ Total Virus Quantity
Up-Spring	4/8	6.67	0.56	4/8	5.77	0.031	2/8	5.34	0.23	10	8.02
Up-Fall	4/8	6.59	1.2	2/8	4.38	0.36	1/8	5.96	na	7	9.15
Low-Spring	2/8	5.87	0.97	1/8	5.83	na	1/8	5.18	na	4	6.89
Low-Fall	5/8	5.22	0.51	3/8	4.99	0.16	1/8	5.87	na	9	7.19
Total fish pos.		15			10			5			
Log_10_ total virus quantity		9.18			6.75			6.36			

^a^ Standard error of the mean (SEM) is reported for virus positive fish within each treatment group, where “na” indicates no SEM was calculated because only one fish was virus positive.

**Table 7 viruses-13-00701-t007:** Prevalence of U and M IHNV infection in spring-run, fall-run, and summer-run Chinook salmon of the CRB during the years 2000–2012.

Adult Migration Timing (Juvenile Life-History)	Total Cohorts Tested ^a,b^	Virus pos. Cohorts	% Virus Pos.	Pos. Cohorts Genotyped	U Pos. Cohorts	% U Pos.	M Pos. Cohorts	% M Pos.
Spring-run								
(stream-type)	777	209	27%	126	103	82%	23	18%
Fall-run								
(ocean-type)	172	43	25%	26	23	88%	3	12%
Summer-run								
(stream/ocean-type) ^c^	62	8	13%	2	1	50%	1	50%

^a^ IHNV Virology, Genotyping and Surveillance (VGS) database records for CRB Chinook salmon 2000–2012 [4]. ^b^ A cohort represents fish of the same species, age (adult or juvenile), collection site location (e.g., Hatchery), and year of testing [4]. ^c^ Summer-run offspring from Chinook salmon populations of the Snake River exhibit a stream-type juvenile life-history, whereas Columbia River populations exhibit an ocean-type life-history [24].

**(a) viruses-13-00701-t008-a:** **Host Population Comparisons.**

Trait	Measure	Variation Observed	Data
*SURVIVAL*		
	*Mortality after high dose virus exposure*	
		Significant difference: Low-Spring is less susceptible to L virus than 3 other host populations.	Figure 2
	*Mortality after moderate dose virus exposure*	
		Significant difference: Low-Spring and Up-Spring are less susceptible to L virus than Up-Fall.	Figure 2
*INFECTION*		
	*Primary data for total # fish infected*	
		Non-statistical observation: Low-Spring has the lowest and Up-Spring the highest # fish infected.	Table 2
	*Primary data for total viral loads*	
		Non-statistical observation: Low-Spring & Low-Fall have higher viral loads than Up-Spring & Up-Fall.	Table 2
	*ID50 values*	
		Significant difference: Low-Spring is less susceptible than Up-Spring for MD virus only.Consistent trends: Low-Spring is also least susceptible to L virus; Up-Spring is most susceptible to UC virus.	Figure 6
	*Infection status on day 3 post-exposure (logistic regression models)*	
		Significant difference at *p* < 0.0083: Low-Spring is less susceptible to infection than Up-Spring (*p* = 0.001).	Table 5
	*Viral load on day 3 post-exposure (generalized linear models)*	
		Significant difference at *p* < 0.0083: Low-Spring has higher viral loads than Up-Spring (*p* = 0.005).Additional trends at *p* < 0.05: Low-Spring has higher viral loads than Up-Fall (*p* = 0.027); Low-Fall has higher viral loads than Up-Spring (*p* = 0.00849); Low-Fall has higher viral loads than Up-Fall (*p* = 0.05)	Table 5
	*Persistence of detectable infection 30 days after virus exposure*	Table 6
		Non-statistical observations: Low-Spring has the smallest number of fish infected; Low-Spring & Low-Fall lower total viral loads than Up-Spring & Up-Fall.	

**(b) viruses-13-00701-t008-b:** **Virus Strain Comparisons.**

Trait	Measure	Variation Observed	Data
*VIRULENCE*		
	*Mortality after high dose virus exposure*	
		Significant differences: L virus causes higher mortality than UC or MD, in all host populations	Figure 2
	*Mortality after moderate dose virus exposure*	
		No significant differences	Figure 2
*INFECTION*		
	*Primary data for total # fish infected*	
		Non-statistical observation: L virus infects more fish than UC or MD	Table 2
	*Primary data for total viral loads*	
		Non-statistical observation: UC virus has higher viral loads than L or MD	Table 2
	*ID_50_ values*	
		Significant difference: L virus is more infectious (lower ID_50_) than UC in Low-OT population only.Consistent trend: L virus is also more infectious (lower ID_50_) than UC in Up-OT population.	Figure 5
	*Infection status on day 3 post-exposure (logistic regression models)*	
		No significant difference at *p* < 0.0166.Trend at *p* < 0.05: L virus is more infectious than UC (*p* = 0.03).	Table 4
	*Viral load on day 3 post-exposure (generalized linear models)*	
		No significant difference at *p* < 0.0166.Trend at *p* < 0.05: UC virus has higher viral load than L (*p* = 0.042).	Table 4
	*Persistence of detectable infection 30 days after virus exposure*	Table 6
		Non-statistical observations: L virus has highest number of detectable persistence infections, MD has the lowest; L virus has higher total viral load than UC or MD.	

## Data Availability

The data presented in this study are available on request from the corresponding author. The data are not publicly available due to fisheries management interests.

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
