# Peer review of "Virulence and Infectivity of UC, MD, and L Strains of Infectious Hematopoietic Necrosis Virus (IHNV) in Four Populations of Columbia River Basin Chinook Salmon"

_viruses, 2021, doi:10.3390/v13040701_

Round 1
Reviewer 1 Report
This work is based on a rather complex infection trial showing infection and mortality from three different genetic strains of IHNV on four different genetic strains of Chinook salmon. The trial hypotheses are grounded on observations of infection status in wild Chinook salmon the Columbia River Basin. The aim of recreating an observation from the wild including several fish populations and IHNV strains experimentally is ambitious, and the article is in general well organized and well written with impressive statistics work, but there are also some obvious concerns.
Main concern:
The aim of trying to test a field observation experimentally is quite ambitious. The hypotheses here are well funded and worth testing, but the scale of the trial ( four fish populations, three IHNV genotypes, many doses, separate mortality/infection experiments) combined with a relatively limited sampling plan and small groups, takes some of the power out of the ambitious aim, and a strong focus on statistic analyses cannot completely save the study. The discussion of many "trends" based on few fish tend to drown the main finding - that the difference in prevalence of infection by IHNV genotypes in different Chinook populations in the wild is not clearly reflected here, and therefore most likely either caused by other factors than viral and fish genetics, or that the trial is not fully representative for the situation in the Columbia River Basin. The study would benefit from clearly discussing some of its limitations. Here, gathering of more information per individual, like monitoring virus levels in fish from the mortality study, more data points from the different infection groups, or using virus isolates from CRB Chinook in the trial could have provided more answers. The authors should address this in general, but particularly based on the specific points below:
1. L144-146, Table 1a, The Chinook populations used in this trial were referred to as previously genotyped populations in Norum et al. 2010 (ref 9), but it is not referred to any control genotyping of eggs or fish for this particular study, confirming that their genetic status is similar to the Norum study. If this is done, this should be clearly stated in the methods section. If not, this should be discussed.
2. L 147, table 1b. The viral strains udsed in this trial appear to be selected based only on 303 bp of the mid-G-region of the IHNV glucoprotein sequence, and apart from that be old isolates (MD from 2007, UC from 1975) originally from steelhead trout, and not directly associated with Chinook from CRB. More relevant and recently collected IHNV isolates would have been more representative, preferrably with the virus genome fully sequenced. This is somewhat touched upon in the discussion (L 716-725), but the choice of isolates should have been discussed more thoroughly in light of how IHNV genetics affects infection and virulence.
3. L166-167 Please clarify the genetic information available on the UC and MD IHNV cases from CRB that the hypothesis was based on. Genotyping based in 300 bp mid-G-region only, or more genetic information available?
4. L198 and L201 Please specify if the eggs or fish were tested negative for IHNV, or just disinfected.
5. L 256-259 Please comment on why IHNV copy number analyses was based on tissue weight/volume homogenate, and not corrected for total RNA concentration. Please give the range in RNA concentrations obtained in the measurements (L256). Did IHNV infection itself affect total RNA?
6. L 285-286 It is stated here that differences between triplicate tanks were not significant. However, MD and UC infection groups only led to 1-2 dead fish overall in three tanks, which seems to be be the actual reason for pooling data. The authors should rather openly present the low fish number behind the mortality curves and not only percentage.
7. L420-422 Authors claim that the Low-spring population is "significantly less susceptible to mortality". It should be noted that this part of the trial lack data to make sure if the dead fish were infected or diseased. The second trial indicate that low spring may be infected to a lower degree. Suggest to change to: "less susceptible to infection or mortality", and comment on the low fish mortality.
8. L 483 In the chapter considering viral doses needed to infect 50% of the population, it should always be specified that this is valid at 3 dpe only. Different genotypes may take longer to infect and pass the epithelium, but still replicate faster when within the host - reaching the main target cells and tissue. This appears to be the situation for the UC strain (low infection rate/high virus levels). The situation at 1, 3 and 6dpe may therefore be completely different, and without time course data, a finding at 3 dpe only give limited information on the actual infection rate.
9. L638: Discussion in general. The authors should avoid overly repeating results, focus on significant findings, and address more of the uncertain and unknown factors from the trial, like unknown virus level and disease state in dead fish from the mortality trial, the infection, replication and eradication time course, and the genetics of the viral isolates. Comparison with similar published trials should be extended to address this.
10. L642-643. The authors should be crystal clear that the viruses used here are not isolates directly from Chinook salmon from the CRB. If any additional information on the genetic relationship to Chinook CRB isolates are available, this should be included and discussed here.
11. L649-650. A design with several doses for infection is great, but at the same time a low number of fish in each group in this trial have complicated the study of viruses with low infectivity and virulence, making differences between the main study groups harder to conclude from. In this case, authors should balance the good intentions with the complex setup, with an honest evaluation of the obvious limitations.
12. L657-658. Hypotheses: The null-hypothesis mentioned here is not completely in line with the purpose of the study from the introduction. The end of the introduction L136-139 clearly points out that the intention is to compare the IHNV UC and MD subgroups in the different populations, with L subgroup as a positive control. Thus, the null hypothesis should be altered to state that there is no difference between the UC and MD subgroups. The main finding is that these groups are not found significantly different under the conditions given, a result that should be considered completely fine, and which is nicely summarized L710-715.
13. L 683, Table 8. A table is useful for the summary of a large and complex trial, but the authors should omit the non-significant trends and observations, and just mention significant findings. Other observations can be discussed in the text of course, but not pointed out in a table when not statistically supported.
14. L819-823: Here, when drawing conclusions from comparison between the infectivity and mortality trial, the authors should also mention that virus analysis and histology from individual fish in the mortality trial in the period of fish death (7-30 dpe) could have provided some valuable additional information. The authors should also discuss the time course/kinetics of infection in other related trials up against the 3 dpe and 30 dpe time points selected here. A different infection curve (different peak point) may lead to wrong conclusions, and different infection curves between U and M viruses have been demonstrated previously (e.g for rainbow trout here: DOI 10.1099/vir.0.012286-0). This should be addressed.
Reviewer 2 Report
I have reviewed the manuscript by Daniel G. Hernandez et al. entitled “Virulence and Infectivity of UC, MD and L Strains of Infectious Hematopoietic Necrosis Virus (IHNV) in Four Populations of Columbia River Basin Chinook Salmon” intended to be published in Pathogens. I find the manuscript suitable for publication after minor revision.
This study is about virulence differences between IHNV genotypes in Chinook salmon populations. This is a well-prepared MS and I recommend the manuscript for publication in Pathogens with minor revision. However, the authors should respond to the comments outlined below.
Line295-298: Cite a paper showing that infection rate at 3dpe can be an indicator of virus infectivity. Line 309: typeo? “dose.p function”
Figure 2: Although Fig 2 is shown as a sum of the three groups, I suggest showing the original data to confirm the reproducibility of infection tests performed with triplicate. Supplemental data is also acceptable. Line 464-466: Please show the result of the significance test. Line 644: "adapted" is not a proper expression. Rather, UC, which is less pathogenic and more proliferative, is considered to be more adapted to Chinook salmon. I think expressions like virulence are better than adapt.
